

# The influence of föhn winds on annual and seasonal surface melt on the Larsen C Ice Shelf, Antarctica

Jenny. V. Turton[1,2,3], Amélie Kirchgaessner[1], Andrew. N. Ross[2], John. C. King[1], Peter Kuipers Munneke[4]

[1]British Antarctic Survey, Cambridge, CB3 0ET, United Kingdom
[2]School of Earth and Environment, University of Leeds, Leeds, LS2 9JT, United Kingdom
[3]Institute of Geography, Friedrich-Alexander University, Erlangen, 91058, Germany
[4]Institute for Marine and Atmospheric Research, Utrecht University, Utrecht, 3508, The Netherlands

*Correspondence to*: Jenny. V. Turton (jenny.turton@fau.de)

**Abstract.** Warm, dry föhn winds are observed over the Larsen C Ice shelf year-round and are thought to contribute to the
continuing weakening and collapse of ice shelves on the eastern Antarctic Peninsula. We use a surface energy balance (SEB) model, driven by observations from two locations on the Larsen C ice shelf and one on the remnants of Larsen B, in combination with output from the Antarctic Mesoscale Prediction System (AMPS), to investigate the year-round impact of föhn winds on the SEB and melt from 2009-2012. Föhn winds have an impact on the individual components of the surface energy balance in all seasons, and lead to an increase in surface melt in spring, summer and autumn up to 100km away from
the foot of the AP. When föhn winds occur in spring they increase surface melt, extend the melt season and increase the number of melt days within a year. Whilst AMPS is able to simulate the percentage of melt days associated with föhn with high skill, it overestimates the total amount of melting during föhn events and non-föhn events. This study extends previous attempts at quantifying the impact of föhn on the Larsen C ice shelf by including a four-year study period and a wider area of interest and provides evidence for föhn-related melting on both Larsen C and Larsen B ice shelves.

**1 Introduction**

In July 2017, an iceberg of approximately 6000 km² calved off the Larsen C Ice Shelf (LCIS) (Hogg and Gudmundsson, 2018), located on the eastern side of the Antarctic Peninsula (AP). Some decades earlier, in 1997 and 2002, the more northerly Larsen A and B ice shelves collapsed almost entirely. This rapid disintegration was preceded by a series of iceberg calving events in previous years, which caused the calving front to recede beyond a compressive arch that provided stability to the ice shelves
(Doake et al, 1998). Since the collapse of Larsen A and B, the rate of discharge of glaciers previously feeding these ice shelves has increased, contributing to loss of land ice (Rignot et al 2004). Therefore, increased attention is now given to observing the response of LCIS to the 2017 calving event, and to anticipating its likely response to future calving events.

Apart from the calving events leading up to the collapse of Larsen A and B, the collapse itself was facilitated by weakening of
the ice due to drainage of surface melt water into crevasses (Scambos, 2002) and increased pressure from ponds of standing



meltwater forming on the ice shelf surface. Surface melt has increased strongly in this region since the middle of the 20[th] century (Cape et al., 2015). As on Larsen A and B (Scambos et al 2000), surface ponding is a common feature on the northwest portion of LCIS (Luckman, 2014), but not elsewhere on the shelf in the current climate. However, surface melt is projected to increase strongly on LCIS in the coming century (Trusel et al., 2018, Bell et al. 2018). For that reason, it is important to

understand the conditions that lead to surface melt, and its future impact on ice shelf stability.

On the ice shelves east of the AP, surface melt is partially caused by föhn winds. These relatively warm and dry winds are caused by westerly air flow over the AP mountain range. Under the right conditions, föhn winds enhance surface melt by providing heat and increased solar radiation to the surface (King et al 2017). As an explanation for the increased surface melt

on Larsen A and B, we can use the hypothesis of Marshall et al (2006), who proposed that a trend towards an increasingly positive Southern Annular Mode (SAM) index in the late 1960's led to a strengthening and southward movement of the circumpolar westerly winds, which increased the flow of air over the AP, and consequently led to an increase in the number of föhn events on the eastern side of the AP. Indeed, a recent study by Cape et al (2015) identified a positive correlation between the SAM and the frequency with which föhn winds were observed over the northern AP for two decades between

1962–1972 and 1999–2010.

Several studies have investigated the origin and characteristics of föhn winds over the LCIS (King et al 2008, Grosvenor et al 2014, Elvidge et al 2015, Turton et al 2018, Wiesenekker et al, 2018, Kirchgaessner et al 2019). Föhn jets are often present during westerly flow, where the föhn air descends through gaps in the topography as well as over the main ridge of the

mountains (Elvidge et al 2015). Depending on whether the westerly flow is linear or non-linear (Elvidge et al 2016), the effect of föhn can be rather localised but intense (non-linear flow), or extensive but weaker (linear flow). Their influence is observable over 100km from the foot of the AP (Kuipers Muneke, 2012., Turton et al 2018). Föhn winds occur year-round and 15 % of the time from 2009-2012 (Turton et al 2018). Wiesenekker et al (2018) found a similar number of 14 % between 2014 and 2016, but also found that their occurrence is highly variable on longer timescales. Also, from season to season, föhn frequency

is highly variable over LCIS: they occur most often during spring, when they can dominate the weather conditions for 65 % of the time (Turton et al 2018). The influence of föhn winds on the surface of the ice shelves along the AP has been a recent focus for modelling and observational studies. Luckman et al (2014) demonstrated that the occurrence of melt ponds over the LCIS relates to the frequency of föhn winds. Similarly, Cape et al (2015) identified strong correlations between the occurrence of föhn events and the frequency of surface melt and the near-surface air temperature in the area of Larsen A and B. Leeson et

al (2017) also suggested that surface melting of Larsen B prior to its collapse was driven by föhn winds.


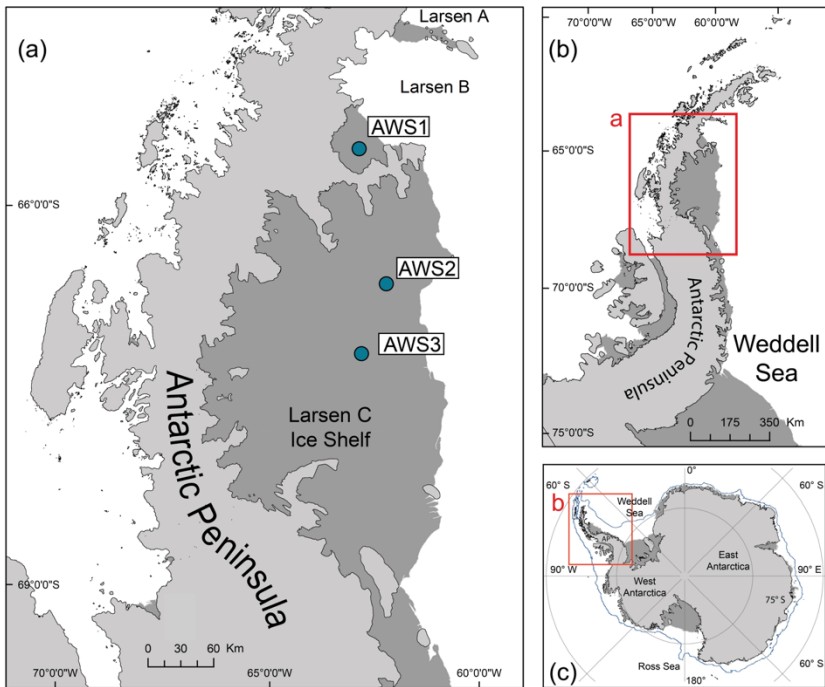

**Figure 1: a) A map of the Larsen C ice shelf and the three AWS locations, b) a map of the Antarctic Peninsula with locations of the other three parts of the Larsen Ice Shelf group, the red outline depicts the area in (a), and the blue line is the average sea ice extent**

**c) A map of the Antarctic continent, with a red outline highlighting the area in (b). This figure is an adapted figure from Turton et al (2018).**

In the summertime surface energy budget (SEB), a positive net radiative flux (fluxes directed towards the surface are defined positive), is only partly offset by negative turbulent fluxes of sensible and latent heat (Van den Broeke 2005). The excess

energy is used for heating and melting of snow. On average, the surface melt rate of the LCIS, derived from satellites and models, is approximately 250 mm w.e yr$^{-1}$, but can exceed 400 mm w.e yr$^{-1}$ in the north (Trusel et al 2013). During winter, heat is extracted from the snow due to net longwave cooling in the absence of solar radiation. A few studies have attempted to quantify the impact of föhn winds on the SEB. However, they have all focused on a number of case studies, on particular seasons with a large number of föhn winds, or on a particular location on the LCIS. For example, Kuipers Munneke et al

(2018) used observations from one location on LCIS, at the foot of the AP mountains to study föhn winds. They identified that 23 % of the annual melt flux at this location was produced during winter (JJA) due to the occurrence of föhn winds. King et al (2017) analysed the effect of föhn winds on the SEB during the 2010-2011 spring melt season using observations and the Antarctic Mesoscale Prediction System (AMPS). They found that, due to föhn winds, melt onset occurred earlier and thus prolonged the melt season in that period. Like other studies using AMPS (Grosvenor et al 2014, Turton et al 2018,



Kirchgaessner et al 2019), King et al (2017) found that AMPS is able to resolve certain characteristics of föhn winds with relative accuracy.

In this study, we aim to extend the scope of previous studies to investigate the relation between föhn and surface melt over the entire Larsen C Ice Shelf and during all seasons, using a continuous time series of observations from Larsen B and C ice

shelves (including previously unpublished data) and high-resolution AMPS output from 2009 to 2012. We assess the impact of föhn winds on the inter- and intra-annual variability in surface melt. By doing so, we investigate the hypothesis that the impact of föhn winds is highest in spring, when föhn winds are more frequent.

## 2 Data

We use three sources of data in this study: automatic weather station (AWS) observations, a SEB model which is driven by

the AWS data, and archived output from the Antarctic Mesoscale Prediction System (AMPS).

## 2.1 AWS observations

Near-surface meteorological data, radiation fluxes, and subsurface temperature were observed at two locations on the LCIS (AWS2 and AWS3) and one (AWS1) on Scar Inlet, a remnant of Larsen B (Table 1 and Figure 1). AWS2 and AWS3 are approximately 100 km away from the foot of the AP Mountains. AWS1 is located further north, and approximately 25 km

away from the AP. Therefore, it observes warmer and more frequent föhn winds (Turton et al 2018). All three stations were located at ~50m a.s.l. when initially erected. Meteorological observations from AWS2 and AWS3 locations were used from January 22 2009 to December 31 2012, and from February 19 2011 to December 31 2012 at AWS1 (Table 1). For this latter period, there is complete coverage by AMPS, the SEB model, and observations. Observations were made every six minutes, from which hourly values were calculated and stored. All AWSs are owned and operated by the Institute for Marine and

Atmospheric Research Utrecht (IMAU) in the Netherlands and maintained by staff of the British Antarctic Survey. For information on instrumentation and sensors, see Kuipers Munneke et al. (2012).

**Table 1: The metadata for the AWS observations. In the current paper, the AWS locations are numbers, following Turton et al (2018). We have also provided the name used in other studies in brackets.**

| AWS number (name) | Coordinates (geographical location) | Meteorological Data availability | SEB model availability |
|---|---|---|---|
| AWS1 (IMAU17) | 65.93°S, 61.85°W (Scar Inlet, Larsen B) | 19/02/2011-31/12/2012 | 19/02/2011- 31/12/2012 |
| AWS2 (IMAU14) | 67.02°S, 61.50°W (Larsen C) | 22/01/2009- 31/12/2012 | 22/01/2009- 31/12/2012 |



| AWS3 (IMAU15) | 67.57°S, 62.15°W | 22/01/2009- 31/12/2012 | 22/01/2009- 26/01/2011 |
|---|---|---|---|
| | (Larsen C) | | |


To compare the relative influence of solar radiation during föhn in different seasons, following King et al (2017) we compute atmospheric transmissivity (tau) as:

$$tau = SW\downarrow / SW\downarrow^{TOP} \qquad\qquad (1)$$

where $SW\downarrow^{TOP}$ is the incident shortwave radiation at the top of the atmosphere. This allows the impact of föhn conditions on

the $SW\downarrow$ to be assessed without seasonal bias due to the extreme changes in potentially available sunlight. $SW\downarrow^{TOP}$ is an output from the SEB model of Kuipers Munneke et al. (2009) outlined in Sect. 2.2.

## 2.2 SEB Model

A previously published and validated SEB model was used to compute the surface energy balance at AWS2 and 3 using the AWS data as input (Kuipers Munneke et al 2009, 2012). Hourly output is available from the SEB model, from which, daily

averages are calculated and used throughout the manuscript. Only a brief overview is of the SEB model provided here, but a detailed description is given in Kuipers Munneke et al (2012). The SEB model is required, as not all components of the SEB (Eq.2) are measured directly by the AWSs. The SEB is here defined as:

$$SW\downarrow + SW\uparrow + LW\downarrow + LW\uparrow + H_{sen} + H_{lat} + G + Q = E \qquad\qquad (2)$$


where $SW\downarrow\uparrow$ are the incoming and outgoing shortwave radiation, $LW\downarrow\uparrow$ are the incoming and outgoing longwave radiation, $H_{sen}$ is the sensible heat flux, $H_{lat}$ is the latent heat flux, G is the ground heat flux, and Q is the amount of shortwave radiation absorbed by the subsurface due to the penetration of the radiation into the snowpack. E is the net energy flux, taking into account the surface and subsurface melting, that is available to heat, melt or cool the ice surface (King et al 2017). We use the

sign convention that all fluxes are positive when directed towards the surface. Therefore a positive E means that the surface is warming and/or melting. To define periods where melt is possible, the following condition is followed:

$$E_{melt} = \{E, T_{SK} = 0 \circ C \qquad\qquad (3)$$
$$\{0, T_{SK} < 0 \circ C$$


The additional term $E_{melt}$ states that melting is possible and is equal to the residual of the SEB calculation (E), when the skin temperature ($T_{SK}$) is at the melting point. Otherwise, the additional energy is not used for melting.

The sensible and latent heat fluxes are calculated using the bulk flux method. The ground heat flux is calculated using a multi-

layer snowpack model, which allows for multiple layers of melting, percolation and refreezing of melt water (Kuipers Munneke



et al. 2012). Within the multi-layer snowpack module, the vertically-integrated change in heat content is calculated to compute the ground heat flux (G). The temperature of the snowpack is initialised using the subsurface temperatures measured by the AWS (at depths of 0.2, 0.3, 0.5, 0.75 and 1.0 m below the surface). Penetration of radiation into the snowpack and the amount of absorbed shortwave radiation (Q) are calculated by a separate module based on Brandt and Warren (1993) and van den

Broeke et al. (2008).

The skin temperature is calculated iteratively, until the SEB is closed (Kuipers Munneke et al., 2012). By Stefan-Boltzmann's law, this skin temperature provides a value of outgoing longwave radiation, which can be compared to the observed flux of longwave radiation for model validation. As the SEB components are derived from a SEB model but based on measurements

by the AWSs, they are referred to as 'observationally-derived' in this paper, to avoid confusing the output with the AMPS data.

### 2.3 Antarctic Mesoscale Prediction System (AMPS)

AMPS is a numerical weather prediction tool that is operationally run by the National Center for Atmospheric Research (NCAR), USA (Powers et al 2012). AMPS is based on the polar version of the Weather Research and Forecasting (WRF)

model and is initiated by Global Forecast System (GFS) data. For the AP domain (domain 6), AMPS output is used here at 5 km horizontal resolution, and 44 vertical terrain-following levels, and at a temporal resolution of 6 hours. Archived output from AMPS is available at various locations, and horizontal-resolutions around the Antarctic (http://www2.mmm.ucar.edu/rt/amps/, last accessed: July 20 2019). For more information on the set-up of AMPS, and how well AMPS resolves near-surface meteorological conditions and föhn winds over the LCIS, see King et al (2015), Turton et al

(2018) and Kirchgaessner et al (2019).

To calculate melt from AMPS data, the Eq. (4) was used:

$$E = SW \downarrow + SW \uparrow + LW \downarrow + LW \uparrow + H_{sen} + H_{lat} \tag{4}$$


where E is the net energy flux available for heating and, potentially, melting the surface when positive. To define periods when melting may occur ($E_{melt}$), the condition outlined in Eq. (3) is used. Following King et al (2015, 2017), G and Q are omitted from Eq. 4, as these are not available in the AMPS output. In AMPS, Q = 0 because no subsurface absorption of solar radiation is taken into account. During melt, G is zero because the temperature gradient near the surface vanishes.

### 2.4 Föhn-Identification

Previously identified föhn winds published in Turton et al (2018) are used. We provide only a brief overview of the method used to identify föhn winds here. Föhn winds were identified from both AWS near-surface observations and upper-air model





output from AMPS, with two different criteria. The method for identifying föhn at the near-surface was based on exceeding thresholds of specific relative humidity values. The absolute relative humidity value used as the threshold depended on the exact location. In the case of slightly more humid conditions, an associated increase in air temperature was also included into the method. The method used to identify föhn from AMPS was based on the height change of a particular isentrope from the windward to leeside of the AP Mountains, in order to isolate the isentropic drawdown which is characteristic of föhn winds over the AP. Only when föhn winds were simultaneously identified in both datasets, was a specific period categorised as Föhn. See Turton et al. (2018) for a discussion on the comparison of Föhn identified by AWS and AMPS.

The term 'föhn conditions' refers to a six-hour averaged period in which föhn winds have been identified. 'Föhn days' are days on which föhn conditions have been identified for at least one six-hour averaged period (but föhn conditions could have been present for the full 24 hours).

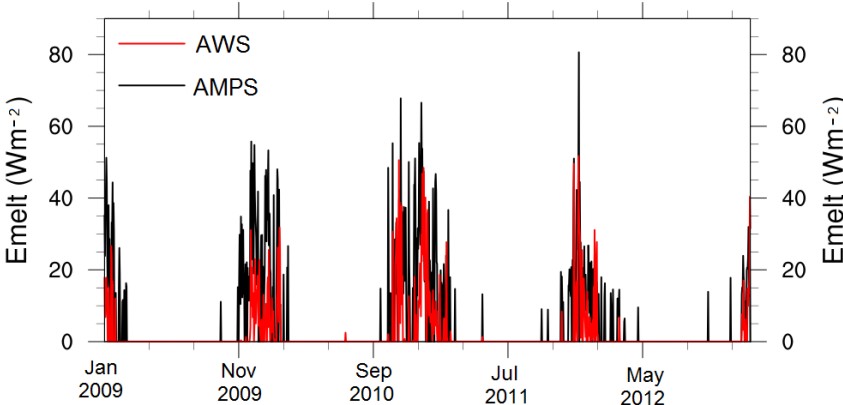

**Figure 2: The daily melt energy $E_{melt}$ observationally-derived at AWS2, and from AMPS data for 2009-2012. Melt estimates by AMPS are plotted in black, observationally-derived melt values are plotted in red.**

## 3 Results

### 3.1 Föhn Conditions

Föhn winds have been observed over the whole Larsen C ice shelf and are most frequent at the foot of the AP Mountains (Elvidge et al 2015, Turton et al, 2018, Wiesenekker, 2018). In particular seasons, föhn conditions can be observed up to 12 % of the time (Table 2). However, it is not uncommon to have a whole season without the occurrence of föhn conditions. They are most frequently observed in spring over the Larsen C ice shelf (AWS2 and AWS3), but closer to the foot of the mountains (AWS1), they are most frequently identified in summer (Table 2). At all three locations in this study, the average six-hourly temperature change associated with föhn events exceeds 11 K, and the relative humidity decreases by at least 19 % (Turton et





al, 2018).Table 2 summarises the percentage of time that föhn conditions were identified at AWS1, AWS2 and AWS3, in each season from 2009-2012.

**Table 2: The percentage of time in which six-hourly föhn conditions were identified from both AWS observations and the AMPS according to Turton et al. (2018). SON is spring, DJF is summer, MAM is autumn and JJA is winter. Refer to table 1 for data gaps**
**which influence the values in DJF, especially for 2009. December values are taken from the preceding year, to keep within the same melting season (for example DJF 2012 is from December 1 2011 to February 29 2012).**

|  | AWS1 [%] | | AWS2 [%] | | | | AWS3 [%] | | | |
|---|---|---|---|---|---|---|---|---|---|---|
| Year | 2011 | 2012 | 2009 | 2010 | 2011 | 2012 | 2009 | 2010 | 2011 | 2012 |
| DJF | 11.1 | 11.5 | 0 | 1 | 1 | 5.2 | 1.4 | 0.6 | 2.2 | 3.8 |
| MAM | 1.4 | 3 | 1.4 | 0 | 1.1 | 1.6 | 4.1 | 0.3 | 1.6 | 4.6 |
| JJA | 2.4 | 3.8 | 1.4 | 4.1 | 0 | 0.8 | 2.2 | 6.0 | 0.5 | 1.1 |
| SON | 7.1 | 7.1 | 2.5 | 9.6 | 1.1 | 3 | 1.9 | 12.1 | 2.5 | 2.7 |

## 3.2 Surface Melting in AMPS

In order to assess AMPS surface melt, we compare it to the observation-derived SEB at AWS2, for which data are available for the full 4-year period. Melt days in both AMPS and SEB are defined as days when melting ($E_{melt}>0$) is observed. During 2009-2012 the SEB model computes 214 melt days at AWS2, where AMPS computes 289 (Table 3). AMPS therefore overestimates the number of melt days compared to observations, likely due to the overestimation of air temperature during non-föhn days in AMPS (Kirchgaessner et al. 2019).


**Table 3: The representation of surface melt from observation-derived data at AWS2 and AMPS data interpolated to the same location. The total number of melt days and percentage of melt days which occur with föhn and non-föhn periods are for 2009-2012. The melt amount values are daily averages over the same period. The total number of föhn and non-föhn days are the same for both AWS and AMPS, as following the föhn identification (Sect. 2.4), föhn conditions must be identified in both to be classified as Föhn.**

| Parameter | AWS2 obs-derived SEB values | AMPS values |
|---|---|---|
| Total number of melt days | 214 | 289 |
| Total number of föhn days | 86 | 86 |
| Number (percentage) of föhn days with melt | 27 (31.4%) | 29 (33.7%) |
| Total number of non-föhn days | 1353 | 1353 |
| Number (percentage) of non-föhn days with melt | 187 (13.8%) | 260 (19.2)% |
| Average $E_{melt}$ (W m$^{-2}$) | 2.0 | 4.9 |





| | | |
|---|---|---|
| Average $E_{melt}$ during föhn days (W m$^{-2}$) | 7.6 | 8.5 |
| Average $E_{melt}$ during non-föhn days (W m$^{-2}$) | 1.6 | 4.7 |


In both the observations and AMPS, over 30 % of föhn days observed at AWS2 lead to surface melt (Table 3). AMPS slightly overestimates the percentage of föhn days which experience melting, but only by 2.3 % (2 days). However, AMPS overestimates the number of melt days coinciding with non-föhn days more considerably (260 non-föhn days experience melting in AMPS compared to 187 in observations). AMPS is therefore able to better represent the occurrence of melting on
föhn days as opposed to melting on non-föhn days.

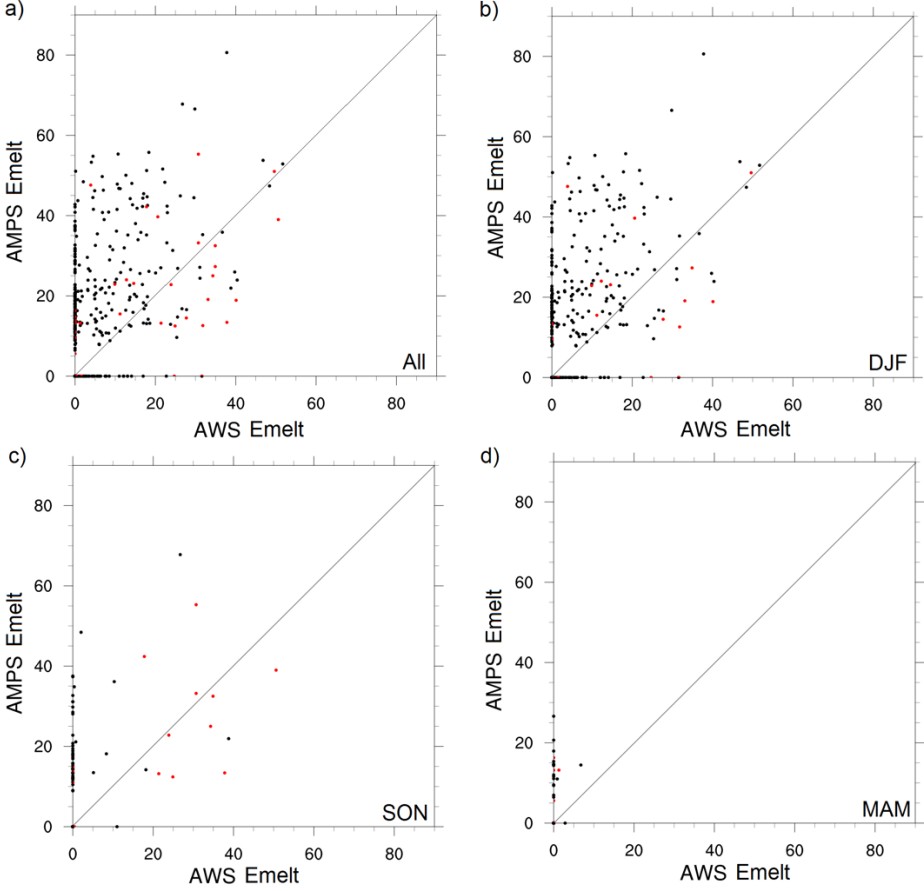

**Figure 3: Scatter plots of daily Emelt values (Wm$^{-2}$) from observations at AWS2 and during a) all months, b) summer months, c) spring months and d) autumn months. Red dots indicate föhn days whilst black dots are non-föhn days.**




Similarly, AMPS overestimates the average $E_{melt}$ during both föhn and non-föhn days (Table 3 and Figure 2). The largest overestimation occurs during non-föhn days, when AMPS simulates a mean $E_{melt}$ of 4.7 W m$^{-2}$ compared to 1.6 W m$^{-2}$ in observationally-derived values at AWS2. However during föhn conditions, $E_{melt}$ is better represented by AMPS, with a smaller positive bias of 0.9 W m$^{-2}$ on average. This can also be seen in Figure 2. AMPS simulates the largest $E_{melt}$ values best, and

overestimates $E_{melt}$ most during non-föhn days. When separating by season (Figure 3), it is clear that AMPS overestimated $E_{melt}$ in all seasons (except winter), but more often during summer. Some of the largest amounts of daily melting often coincide with föhn days during spring and summer (Figure 3b,c), and AMPS is better able to represent events with large melt amounts than those with low melt rates. These results agree with findings by Kirchgaessner et al. (2019), who found that AMPS underestimates the air temperature during föhn days but overestimates the temperature the rest of the time, which leads to a

higher surface temperature and a higher likelihood of melting on non-föhn days (Kirchgaessner et al. 2019). Figure 3c also highlights that the majority of melting during spring is associated with föhn days, which AMPS represents well.

The downwelling shortwave radiation is overestimated by AMPS for this location (King et al 2015, 2017). Combined with the low albedo value and the poor representation of clouds in AMPS, the ice surface is much warmer in AMPS than in reality

(mean bias of 1.8 K). Therefore, on days in late spring and summer when the skin temperature is close to the melting point in observations, AMPS will simulate that it is already at 0 °C, which leads to the overestimation in the total number of days with melting (Figure 3a,b). The overestimation of melt days was also found in other studies using AMPS and the polar WRF model over ice shelves (Grosvenor et al. 2014; King et al., 2008, 2015, Kirchgaessner et al 2019).

During föhn conditions, when the melt amount is better represented by AMPS, there are a number of reasons for the reduced overestimation of $E_{melt}$, in conjunction with the lower air temperatures identified by Kirchgaessner et al (2019). AMPS overestimates the latent heat flux ($H_{lat}$). The observation-derived values of $H_{lat}$ at AWS1, AWS2 and AWS3 during föhn composites are -6.2 W m$^{-2}$, -3.6 W m$^{-2}$ and -1.3 W m$^{-2}$ respectively (Table 4). AMPS simulated $H_{lat}$ values of -24.4, -10.6 and -10.9 W m$^{-2}$ for AWS1, AWS2 and AWS3 locations respectively, during föhn conditions. AMPS also overestimates the net

longwave flux ($LW_{net}$) during föhn conditions, but with much smaller biases than for $H_{lat}$. During föhn conditions, AMPS simulates an average $LW_{net}$ of -36.1 W m$^{-2}$ for AWS1, -43.0 W m$^{-2}$ for AWS2 and -43.7 W m$^{-2}$ for AWS3. From the observation-derived SEB values, AWS1, AWS2 and AWS3 have $Lw_{net}$ values of -34.8, -40.6 and -41.1 W m$^{-2}$ during föhn conditions respectively. A combination of lower (more negative) $LW_{net}$ and $H_{lat}$ during föhn days in the simulations acts to cool the surface more than is observed, which could be responsible for the better representation of $E_{melt}$ during föhn days

compared to non-föhn days in AMPS.

Regardless of the overestimation of non-föhn melt in AMPS, it is evident from the observations and AMPS, that melting during föhn conditions is significantly higher (at the 99 % confidence level) during föhn conditions than during non-föhn conditions, even more than 100 km away from the foot of the AP mountains. As AMPS is able to reproduce föhn-related melting, we have





used it to assess the spatial distribution of föhn-induced melting for the entire ice shelf (Figure 4). Föhn-induced melting is most frequent in the north of the ice shelf, largely mirroring the spatial distribution of föhn conditions and the near-surface air temperature (Turton et al. 2018). During summer, föhn-induced melting is simulated over the whole area from Scar Inlet in the north across the entire Larsen C ice shelf to 70°S. Outside of summer, föhn-induced melting is still prevalent over the ice shelf, with over 40 melt events (six-hourly) simulated on Scar Inlet during spring (Figure 4b). There are more föhn-induced melt events during spring than in autumn, likely related to the higher occurrence of föhn during spring.

The following analysis, separated into annual, interannual and seasonal impact of föhn, uses the observations and derived SEB components at the three locations to quantify the impact of föhn on the ice shelf.

### 3.3 Annually-averaged impact of föhn

Table 4 presents the annual-averaged differences between föhn and non-föhn conditions in some of the observed SEB components. In the SEB data, 14 % of non-föhn days at AWS2 from 2009-2012 were melt days. These are largely confined to summer months, when melting occurs annually. The frequency of melt days more than doubles when assessing föhn days, with 31 % of föhn days at AWS2 coinciding with melt days. A similar magnitude of increase was observed at AWS3, where melt-day occurrence increased from 12 % during non-föhn conditions to 20 % during föhn days. Therefore, even at a distance of 100 km from the foot of the mountains, föhn conditions are able to increase the number of melt days per year. The largest increase in melt-day occurrence was at AWS1, where the percentage of days observing melt increased from 14 % during non-föhn days, to 43 % during föhn days in 2011 and 2012.

During föhn conditions incoming shortwave radiation (SW↓) is hypothesized to be larger due to the clearance of clouds, which can occur in the lee of the AP mountains (Grosvenor et al 2014, Elvidge and Renfrew, 2015). However, due to the large annual cycle in SW↓ in the Polar Regions, this could bias the difference between föhn and non-föhn days if they are not evenly distributed throughout the year. From Table 2 and Turton et al (2018), it is evident that the föhn days are not evenly distributed. Therefore, the shortwave transmissivity (tau) (see Data and Methods) is a more reliable indication of the impact of föhn winds on the downwelling shortwave radiation. Data show an increase in shortwave transmissivity at all three locations during föhn conditions, indicating an increase in the incoming shortwave radiation due to cloud clearance (Table 4 and Figure 5d). However, the differences in tau between föhn and non- föhn conditions are small and are not statistically significant.





**Table 4: Annual average values for observed SEB components and surface temperature during composites of föhn and non-föhn periods at AWS1, AWS2 and AWS3. * indicates a statistically significant difference between non-föhn and föhn values at 95 % confidence level using the t-test, and ** 99% confidence. This is not tested for tau, as this is calculated from the average SW and SW$_{top}$, and therefore the sample size is too small.**

| Variable | Non-föhn Periods | | | Föhn Periods | | |
|---|---|---|---|---|---|---|
| | AWS1 | AWS2 | AWS3 | AWS1 | AWS2 | AWS3 |
| SWnet (W m-2) | 18.3 | 15.7 | 11.7 | 30.4** | 31.1** | 18.3** |
| tau | 0.61 | 0.58 | 0.58 | 0.65 | 0.67 | 0.65 |
| LWnet (W m-2) | -13.7 | -13.9 | -12.1 | -34.8** | -40.6** | -41.1** |
| Hsen (W m-2) | -3.4 | -0.5 | 1.0 | 19.7** | 23.0** | 24.2** |
| Hlat (W m-2) | -3.4 | -2.5 | -1.6 | -6.2** | -3.6 | 1.3 |
| Emelt (W m-2) | 2.4 | 1.6 | 0.4 | 10.0** | 7.6** | 1.8* |
| Tsk (°C) | -16.8 | -16.8 | -17.7 | -7.9** | -8.3** | -9.9** |

The sensible heat flux (H$_{sen}$) was much larger (and positive) during föhn conditions than during non-föhn conditions (Table 4, Figure 5). This is due to the increased air temperature and higher wind speed, both leading to an increased supply of heat to the surface. The annual average sensible heat values observationally derived during föhn days for AWS2 and AWS3 are very similar (23.0 W m$^{-2}$ and 24.2 W m$^{-2}$ respectively) (Table 4). However, at AWS1 it is slightly smaller (19.7 W m$^{-2}$) due to the slightly lower wind speeds under föhn conditions at this location compared to the other locations. In most locations along the foot of the AP, the wind speeds are higher than further downstream. However, for the two years of available data at AWS1, this is not the case here. Despite the smaller H$_{sen}$ value at AWS1, the increase in H$_{sen}$ between non-föhn and föhn days is similar to the other locations; between 23.1 W m$^{-2}$ at AWS1 and 23.5 W m$^{-2}$ at AWS2. Figure 5a highlights the significant increase in H$_{sen}$ during föhn. The high H$_{sen}$ during föhn conditions impacts the whole of the Larsen ice shelves, with 25 W m$^{-2}$ values simulated over all of Larsen B and C ice shelves (Figure 6d). Very close to the foot of the AP mountains, AMPS simulates higher Hsen values, where föhn conditions are stronger in the inlets, and föhn-induced melting can also occur in winter (Kuipers Muneke et al 2018).


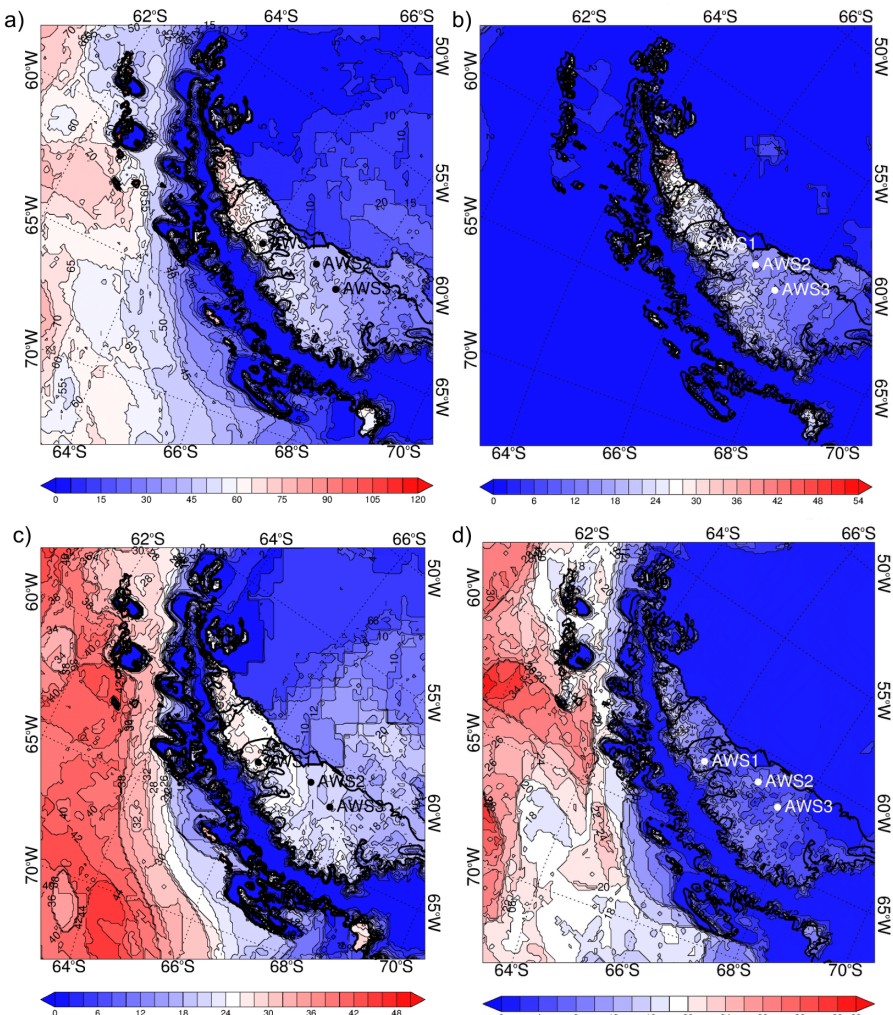

**Figure 4: The number of 6-hourly melt events in AMPS during all föhn events (a), spring (b), summer (c) and autumn (d). Winter is not included as there was no melt days simulated by AMPS.**

The latent heat flux becomes more negative during föhn days at AWS1 and AWS2, which we can attribute to the increase in sublimation of surface snow increases due to the dry air over the ice shelf (Figure 5). At AWS3, there is an increase in $H_{lat}$ (1.3 W m$^{-2}$) during föhn conditions but this difference to non-föhn conditions is not statistically significant. At AWS1, the difference in $H_{lat}$ between föhn and non-föhn conditions is larger than at the other locations and is significant. This is likely due to the drier föhn conditions observed close to the AP, whereas at about 100 km distance, the air has become more moist due to mixing with pre-existing non-föhn air, and therefore less sublimation occurs. This is evident in the AMPS simulation (Figure 6), with more negative values of $H_{lat}$ closer to the AP during föhn days.



$LW_{net}$ becomes significantly more negative during föhn events (Figure 5), which is likely related to the 'föhn clearance', whereby there are fewer clouds during föhn conditions which reduces the downwelling flux of longwave radiation (Elvidge and Renfrew, 2016). Simultaneously, the warmer surface increases the outgoing longwave radiation flux, which both

contribute to a more negative $LW_{net}$ than during non-föhn conditions. Despite the larger negative fluxes of $LW_{net}$ and $H_{lat}$ during föhn conditions, the positive increase in $SW_{net}$ and $H_{sen}$ contributes to more energy being available for melt ($E_{melt}$) during föhn conditions.

$E_{melt}$ more than tripled during föhn conditions compared to non-föhn conditions, from 1.6 to 7.6 W m$^{-2}$ at AWS2, and from 2.4

to 10.0 W m$^{-2}$ at AWS1 (Table 4). AWS1 is closer to the foot of the AP Mountains and therefore experiences warmer and more frequent föhn conditions, which contributed to the significantly higher amount of energy available for melt during föhn periods (Table 4). Figure 6 presents a spatial distribution of the energy flux during föhn periods in AMPS. A higher energy flux is present over Larsen B than Larsen C during föhn winds. Therefore, as well as additional melt days, the amount of melting on those days also increases in association with föhn winds. The combination of these generates föhn-induced melting

of the ice shelf.

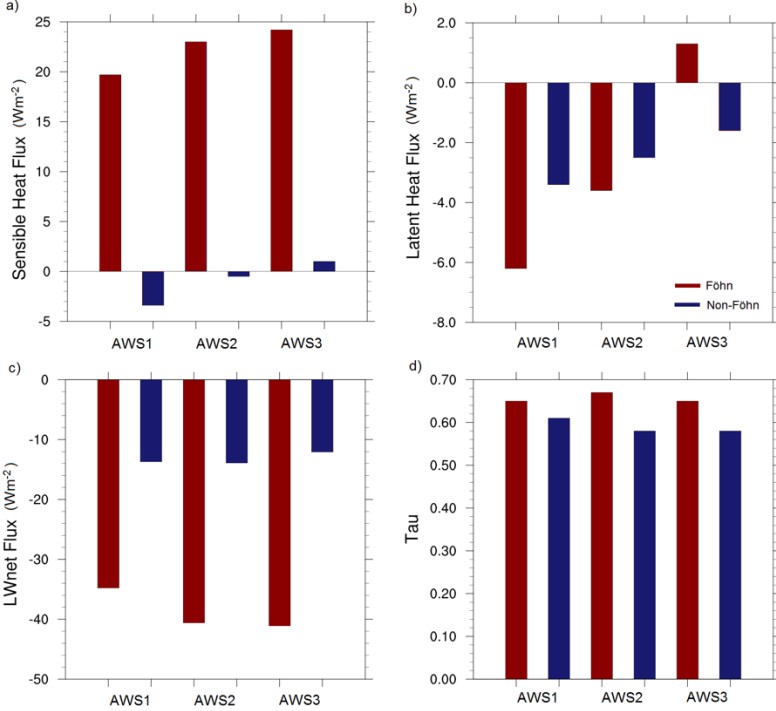

**Figure 5: The sensible heat flux (a), latent heat flux (b), net longwave radiation flux (c) and tau (d) from the three locations during föhn (red) and non-föhn (blue) periods.**






### 3.4 Interannual melt

Due to the poor availability of data at AWS1 and AWS3 locations, this section will focus solely on AWS2 data. The annual average amount of melt from 2009-2012 (föhn and non-föhn conditions combined) was 180 mm w.e. yr$^{-1}$ at AWS2. Excluding föhn days, the annual average melt amount at AWS2 reduced significantly to 146 mm w.e. yr$^{-1}$. In other words, föhn increases

the melt volume at AWS2 by 34 mm w.e. yr$^{-1}$ (18 %). The majority of non-föhn melting is restricted to the summer months. Therefore, the potential for föhn winds to create additional melting, outside of the usual melt season is of interest, as they could extend the melt season.

The influence of föhn winds on surface melt is largest in the years with a large number of föhn conditions during the early

spring-summer period (September-December). The year 2010 was exceptional in this regard, with 35 six-hourly föhn conditions identified at AWS2 alone. For comparison, the number of identified föhn conditions at AWS2 from September to December in 2009, 2011 and 2012 were 12, 13 and 11 respectively. The annual melt amount at AWS2 in 2010 was 258 mm w.e. This annual total decreases by 76 mm w.e to 182 mm w.e. when the melt associated with föhn conditions is removed. Focusing specifically on spring (SON) 2010: there were 22 föhn days, of which 11 of them generated melting (50 %) at AWS2.

Conversely, there were 69 non-föhn days in spring 2010 (AWS2), of which just 8 days experienced melting (12 %). King et al. (2017) studied the impact of the föhn conditions in November 2010 in more detail and identified that the duration and frequency of melting over the LCIS was increased due to föhn winds in this period.

In contrast to 2010, only 11 föhn conditions were identified at AWS2 from September to December in 2012. The annual melt

amount in 2012 was 83 mm (significantly less than in 2010 at the 95% level). When föhn days were removed from the 2012 analysis, the annual total melt only decreased by 0.1 mm w.e. During spring, little melt is observed except on föhn days. Hence, the frequency of föhn winds in spring has an impact on the melt.

The annual number of melt days, energy available for melt, annual melt amount and length of the melt season all increased

due to the occurrence of föhn winds, especially in years when a large number of föhn conditions were identified during the extended summer period (Oct-Mar). We will now present the impact of föhn winds in separate seasons, but with a particular focus on spring.

### 3.5 Spring

The largest increase in surface melting is experienced during föhn winds in spring (SON). Although not all changes in mean

values between non-föhn and föhn conditions were statistically significant, the changes in the SEB indicate a large impact due to föhn winds. Table 5 displays the average values for composites of föhn and non-föhn periods during spring.





**Table 5: Spring daily averages of surface temperature and SEB components for föhn and non-föhn conditions. The \* indicates statistical significant difference between föhn and non-föhn periods using the T-test at 95% level and \*\* is at the 99% confidence**
**level.**

| Variable | Non-föhn Periods | | | Föhn Periods | | |
|---|---|---|---|---|---|---|
| | AWS1 | AWS2 | AWS3 | AWS1 | AWS2 | AWS3 |
| SWnet (W m-2) | 27.8 | 23.4 | 18.4 | 28.2** | 36.3** | 25.9* |
| tau | 0.57 | 0.60 | 0.60 | 0.54 | 0.68 | 0.66 |
| LWnet (W m-2) | -19.7 | -22.6 | -18.8 | -37.7* | -43.5** | -45.2** |
| Hsen (W m-2) | -5.3 | 3.5 | 2.2 | 22.2** | 25.4** | 23.9** |
| Hlat (W m-2) | -4.2 | -2.5 | -1.8 | -6.0** | -5.4* | -0.1 |
| Emelt (W m-2) | 0.3 | 1.2 | 0.3 | 3.8** | 7.7** | 2.1* |
| Tsk (°C) | -16.0 | -16.4 | -15.1 | -9.1** | -7.5** | -7.0** |

The atmospheric transmissivity (tau) increased during föhn conditions indicating an increase in incoming shortwave radiation at the surface, although not statistically-significantly. The net longwave radiation was significantly (95 % confidence level)
lower during föhn conditions than during non-föhn conditions at all locations (Table 5). Both turbulent fluxes exhibited significant differences during föhn conditions compared to non-föhn conditions at AWS1 and AWS2. The sensible heat flux increased (more positive) by over 20 W m$^{-2}$ at all locations (Table 5), and the largest increase was observed at AWS1 (27.5 W m$^{-2}$). The warmer air and higher wind speeds contribute significantly to increasing the H$_{sen}$ over the ice shelf. The latent heat flux is more negative at AWS1 and AWS2 during föhn conditions, indicative of sublimation and evaporation. However,
at AWS3 H$_{lat}$ increased, although this was not statistically significant (95 % confidence interval).

The cooling effect of the net longwave radiation and latent heat flux was not large enough to counteract the considerable heating processes, therefore, melt energy was available during spring föhn conditions. At AWS2 during spring, melting occurred on just 3 % of non-föhn days, due to the low temperatures in the absence of föhn winds. However, melting increases
significantly when accounting for föhn conditions. In spring, 28 % of föhn days coincided with observed melting. A similar increase was found during föhn days at the other locations: At AWS1 just 5 % of non-föhn days coincided with melt days, whilst 30 % of föhn-days coincided with melt days. At AWS3, 4 % of non-föhn days and 28 % of föhn days experienced melting. Föhn winds therefore contribute to an increase in the number of melting days per year at all observed locations, including those at over 100 km from the foot of the AP.






At AWS2, the average energy available for melt ($E_{melt}$) during spring föhn conditions was 7.7 W m$^{-2}$ (Figure 7). This was greater than the mean daily melt energy during summer at this location (7.0 W m$^{-2}$). The amount of melt energy associated with föhn conditions at AWS1 was lower (3.8 W m$^{-2}$) than at AWS2, however this does not take into account the large melt amount and early melt onset associated with föhn winds in spring 2010, as data are only available from February 2011 to December 2012. When assessing the annual average melt energy for 2012 (period in which observations overlap), there is considerably more daily melt energy at AWS1 (3.5 W m$^{-2}$) than at AWS2 (1.0 W m$^{-2}$).

Therefore, during spring, föhn conditions increase both the average rate of melt production, and the number of melt days, both close to the foot of the AP and up to 130km away.

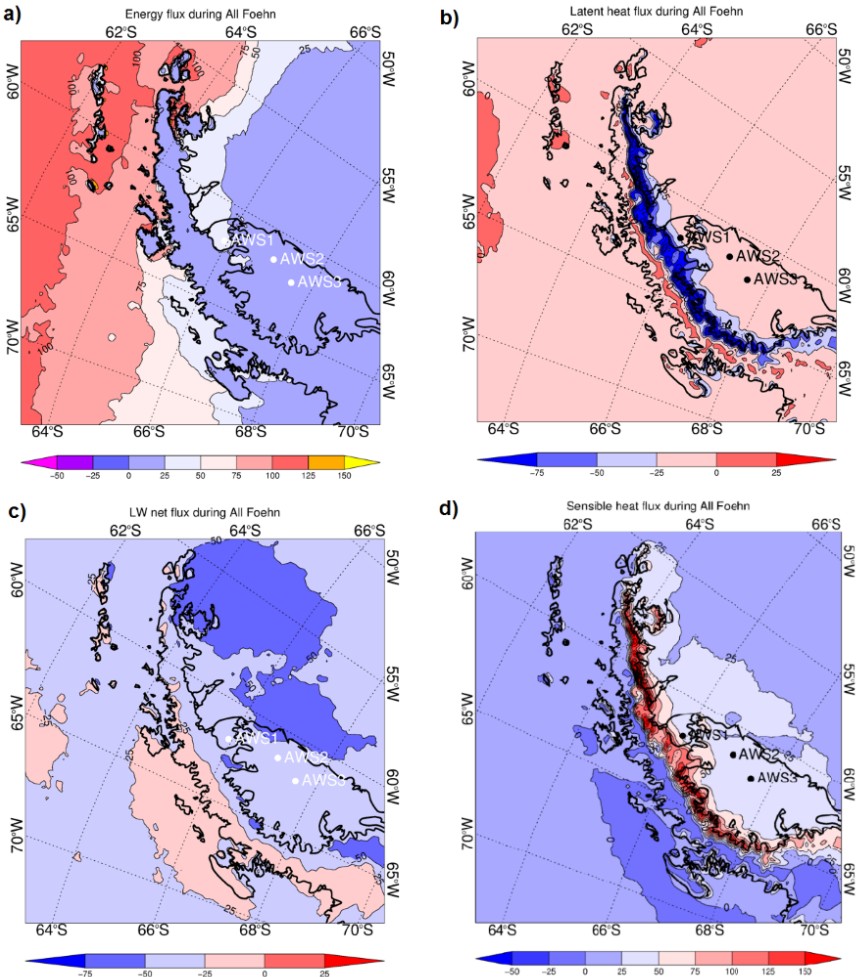

**Figure 6: The energy (a), latent heat (b), net longwave (c) and sensible heat (d) fluxes as a composite of all föhn events from 2009-2012 in AMPS.**





## 3.6 Summer

The energy available for melt and percentage of melt days during summer is relatively high, regardless of additional föhn induced melting, due to higher air temperatures and larger SW↓ during this season (Figure 7, Table 7). A day may already have experienced melting, and the presence of föhn winds was coincidental and did not cause the melting. However, from previous studies, it has been found that individual föhn events can increase or prolong melt when it occurs during summer (Elvidge et al.,2016; Kuipers Munneke et al., 2012).

**Table 6: Summer daily average values of SEB components and surface temperature during composites of föhn and non-föhn conditions at AWS1, 2 and 3. The * indicates statistical significant difference between föhn and non-föhn periods using the T-test at 95% level and ** is at the 99% confidence level.**

| Variable | Non-föhn Periods | | | Föhn Periods | | |
|---|---|---|---|---|---|---|
| | AWS1 | AWS2 | AWS3 | AWS1 | AWS2 | AWS3 |
| SWnet (W m-2) | 46.1 | 38.0 | 27.8 | 63.4** | 57.9** | 50.0** |
| tau | 0.61 | 0.61 | 0.59 | 0.66 | 0.7 | 0.68 |
| LWnet (W m-2) | -23.4 | -22.7 | -18.9 | -39.9* | -45.4** | -45.3** |
| Hsen (W m-2) | -8.2 | -4.2 | -2.3 | 12.3** | 8.7** | 6.0** |
| Hlat (W m-2) | -9.8 | -8.5 | -6.2 | -12.8* | -7.9 | -5.5 |
| Emelt (W m-2) | 12.1 | 7.0 | 1.6 | 30.8** | 18.3** | 6.0* |
| Tsk (°C) | -3.7 | -3.1 | -4.8 | -2.0** | -3.6 | -2.9 |

There was a significant increase in the net shortwave radiation and a decrease in net longwave radiation during summer föhn periods in comparison to non-Föhn conditions, likely, due to the cloud clearing during föhn (Table 6) (Grosvenor et al 2014). The sensible heat flux significantly increased at all three locations during föhn conditions, which changed the direction of energy transport from negative (away from the surface) during non-föhn to positive (downwards) during föhn conditions. Negative sensible heat flux, associated with convection, is common at AWS2 during summer (Kuipers Munneke et al., 2012).

As a consequence of the higher surface temperatures, sublimation and evaporation are common in summer, leading on average to a negative latent heat flux. Under föhn conditions, the change in conditions was mixed. At AWS1, the latent heat flux became more negative during föhn conditions, indicative of enhanced sublimation (Table 6). At AWS2 and AWS3, $H_{lat}$ increased, however the changes from non-föhn to föhn were not significant.





The increase in $E_{melt}$ during föhn conditions was statistically significant at all locations and was largest at AWS 2 (95 % confidence level), increasing from 7.0 to 18.3 W m$^{-2}$. During 78 % of föhn days the surface was melting, compared to just 54

% of non-föhn summer days (at AWS2). The number of melt days and the melt energy both increase in summer under föhn conditions. Despite the already warm conditions and high melt amount, there are statistically significant changes to the SEB components and melt energy due to föhn conditions in summer.

**Table 7: Melt onset and end dates. Melt onset refers to the first day with observed melting. Melt end is the last day of the melt season**

**on which melt is observed. Dates are taken from AWS2 to allow comparison with Luckman et al. (2014) melt onset and end dates.**
**A range of values are given due to the uncertainty of reading Figure 2 from Luckman et al. (2014) and are not the variation in dates.**

| Year | Melt Onset | Melt end |
|---|---|---|
| Approx values from 2006-2012. From Luckman et al. (2014) | January 3 to January 13 | February 23 to March 3 |
| 2009 | November 23 2009 (non-föhn) | February 19 2010 (non föhn) |
| 2010 | October 27 2010 (föhn) | May 15 2011 (föhn) |
| 2011 | November 8 2011 (föhn) | January 27 2012 (non-föhn) |
| 2012 | December 13 (non-föhn) | After observation period |

Luckman et al. (2014) presented the average melt onset and end dates from 2006 to 2012 for the Larsen C ice shelf, taken from

satellite radar backscatter observations. At the location of AWS2, the approximate start and end dates of the melting season are provided in Table 7. The earliest onset of a melting season (in 2009-2012) was on October 27, 2010, which was associated with a föhn event. Similarly, the end of the melt season was associated with föhn-induced melting observed on May 15, 2011. This is far outside of the typical melt season at this location. The preceding melt day was on March 5, 2011 and was not associated with föhn winds. Therefore, föhn winds have the ability to induce melting outside of the usual summer melt period.

**3.7 Autumn**

The sensible heat flux was positive and significantly larger during föhn days in all three locations, and in AMPS output (not shown). On average, $H_{sen}$ increased from 0.0 W m$^{-2}$ during non-föhn periods to 20.9 W m$^{-2}$ during föhn conditions at AWS2. This agrees well with the AMPS output, which simulates a mean $H_{sen}$ of 24.9 W m$^{-2}$ during föhn days in autumn at AWS2. Surface warming was also significantly larger during föhn days than during non-föhn days, raising the surface temperature by

11.8 K at AWS2.

Figure 3d highlights the low daily $E_{melt}$ values at AWS2 during autumn. The energy available for melt only increased by 0.1 W m$^{-2}$ (at AWS2) between föhn and non-föhn days. Melting only occurs on 1 % of non-föhn days, whereas 8 % of föhn days experience melting at AWS2. Closer to the AP, föhn-induced melting was higher than at the stations further east. The

percentage of föhn days experiencing melt at AWS1 was 43 % compared to just 1 % of non-föhn days. Daily average $E_{melt}$



increased from 0 W m$^{-2}$ during non-föhn days to 4.3 W m$^{-2}$ during föhn days. Therefore, föhn-induced melting is possible during autumn, although it is limited in extent, and only occurs very close to foot of the AP mountains, where föhn winds are warmer.

### 3.8 Winter

The smallest impact from föhn conditions on surface melt was observed during winter. The radiation deficit in winter, mainly due to the lack of solar radiation, is so large that increased sensible heat during föhn can almost never bring the surface to the melting point, except in the inlets of the AP (Kuipers Munneke et al 2018). There was no melting observed at any AWS location (or anywhere on the ice shelf in the AMPS output) during winter between 2009 and 2012. Nonetheless, föhn did have an impact on the individual SEB components during winter, as the air temperature can often rise above freezing (Kirchgaessner

et al 2019).

$H_{sen}$ and $H_{lat}$ both experience a significant change between föhn and non-föhn days. At AWS2 the average $H_{sen}$ increased from 3.4 W m$^{-2}$ to 36.6 W m$^{-2}$ during föhn days, and the latent heat increased from 0.6 W m$^{-2}$ to 2.9 W m$^{-2}$. A similar magnitude of change in $H_{sen}$ and $H_{lat}$ was observationally-derived at AWS3 and AWS1. The large increase in $H_{sen}$ is attributed to the

considerably warmer (and often windier) conditions during föhn. There was no observation-derived melting during winter föhn events.

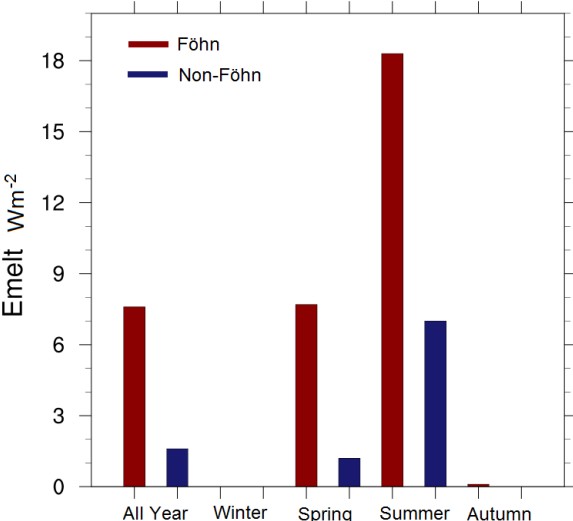

**Figure 7: The observationally-derived melt energy E$_{melt}$ values from AWS2 during föhn and non-föhn conditions, separated by seasons.**




## 4 Discussion

This study uses three locations for SEB calculation from observations, to provide a larger spatial interpretation of melting associated with föhn winds. Despite that the low number of observations is still a limitation of this study. The most reliable SEB dataset was obtained for AWS2 where a SEB model was run by Kuipers Munneke for 2009 to 2012. Unfortunately, no SEB data were available for this entire period at the foot of the AP (AWS1), where the largest melt rate and highest number of melt days have been previously observed in satellite images (Luckman et al., 2014). More recently, data have become available for more locations on the LCIS, including the inlets, which observe high melt rates associated with föhn winds (Kuipers Munneke et al. 2018). Here, observations and the SEB model were available from February 2011 to December 2012 and have shown evidence of föhn-induced melting close to the AP. This is the first time that the results from the SEB model have been analysed and presented for Larsen B (AWS1).

The large increase in melt energy, number of melt days and increased duration of melting during spring are likely the biggest impacts of föhn conditions on the surface of the Larsen Ice Shelf. This föhn-induced melting at AWS2 during spring is comparable in energy and melt amount to that of non-föhn conditions in summer. In the absence of föhn conditions in spring, the melt amount is significantly smaller. In one particular melt season of 2010/2011, the melt season was 164 days long due to an early onset on October 27, 2010 and a late end date of May 15, 2011. This is a much longer melt period than has been observed at this location between 2009-2012. Luckman et al (2014) published melt duration, melt onset and melt end dates for the LCIS from 2006-2012, and showed the average melt duration for the LCIS is 50-60 days. Therefore, föhn winds can significantly extend the melt duration.

During the period under investigation in this study there were no melt days in either föhn or non-föhn days during winter at the three AWS locations used here. There is a high spatial and temporal variability in the occurrence and strength of föhn winds over the LCIS (Turton et al. 2018), which likely explains the contradicting results to those by Kuipers Munneke et al. (2018), who identified high rates of winter melting associated with föhn winds. The winter melting period investigated previously was not within the period investigated here. With a longer observational period at AWS1 now available, this site should be investigated further, as winter melt may be specific to individual years.

The number of melt days estimated by AMPS exceeded the observationally-derived values, especially during non-föhn days. One reason for the melt day overestimation is the positive bias in near-surface temperature during non-föhn conditions in AMPS (Kirchgaessner et al 2019). This is caused by the positive bias in incoming shortwave radiation, which results from the poor representation of clouds in the model (Listowski et al. 2017), together with the low albedo value used in AMPS. This has been discussed by Grosvenor et al. (2014); King et al. (2015) and King et al. (2017). Therefore currently, the values of melt energy from AMPS cannot be trusted if used as an absolute estimate of melting specifically caused by föhn winds. However,



it can be used to infer the spatial patterns of melting during föhn days, and the average melt energy when including all melt
515    events. The poor representation of clouds in many regional climate models causes issues in the accuracy of SEB and melt
information. Recently, Gilbert et al. (2020) found that cloud-phase during the austral summer strongly influences the amount
of melting on the LCIS and can determine whether melting is simulated or not by the MetUM model.

Satellites can provide a longer-term perspective to the period 2009-2012 that we study here. According to QuikSCAT, the
520    number of melt days was particularly low in the 2009-2010 and 2012-2013 summer seasons (Bevan et al 2018). The 2010-
2011 and 2011-2012 summer seasons have a longer melt periods, however in the context of the last 18 years, they were not
considered high-melt years (Bevan et al 2018). The spatial pattern of melting in these two seasons had a bimodal melting
pattern, where the northern section of the LCIS had a higher number of melt days (60-75 days) than the southern part of the
LCIS (10-25 melt days) (Bevan et al 2018). Whilst the three observational sites used in the current study are in the higher-melt
zone, there were relatively fewer föhn events in the 2009-2012 period than during other periods between 1999 and 2017 used
in Bevan et al (2018). Therefore, the effects of föhn winds on the SEB could be even greater than we have highlighted in this
study, if the SEB could be calculated for particularly high-melt years.

## 5 Conclusions

The discrimination between föhn and non-föhn conditions provides a robust understanding of the impact of föhn on
components of the SEB and ultimately, surface melt, by assessing the more general response to föhn, as opposed to studies of
individual events. The limitation of assessing case studies is that the chosen event may be an anomaly, or not representative of
the average föhn conditions, whereas assessing the average impact of föhn provides more confidence in the quantification of
surface melt due to föhn.

This study comes to similar conclusions as studies by King et al. (2015) and King et al (2017). Especially in spring, föhn
conditions have the potential to prolong the melt season by initialising an early onset of the melt season. Moreover, this study
concludes that the intensity of melt increases during föhn conditions, even 100km from the AP. Föhn conditions have a large
impact on the sensible heat flux, which leads to an excess of energy that is available to heat and melt the snow during spring,
summer and occasionally, autumn.


The results presented here and in previous studies (Bevan et al 2018, Luckman et al 2014, Wiesenekker et al, 2018) highlight
the large interannual variability in melt amount and duration. Similarly, there is a large interannual variability in the spatial
and temporal distribution of föhn winds over LCIS. The number of melt days over the majority of the LCIS has decreased
between the years 2000 and 2016. However, close to the foot of the AP, where föhn winds are strongest and most frequent,
the amount of melt has increased over this same time period (Bevan et al 2018). Regional climate models such as the MetUM
(Elvidge et al 2015), WRF model (Turton et al 2017) and RACMO2 (Weisenekker et al 2018) can now accurately represent



near-surface conditions during föhn over the LCIS and these could be used to extend the study in future research. However, AMPS struggles to represent melting outside of föhn conditions, and although it performs better during föhn conditions, this is likely due to the higher negative biases in net longwave radiation and latent heat flux, and not due to a good representation

of föhn warming. Therefore, to completely trust the impact of föhn winds on the SEB of the Larsen C ice shelf, the components of the SEB should be improved in AMPS.

**Data Availability**

Subsets of the AMPS output are available online at https://www2.mmm.ucar.edu/rt/amps/. The surface energy balance values

from AWS1, 2 and 3 are available upon request from Peter Kuipers Munneke at P.KuipersMunneke@uu.nl.

**Author Contribution**

Analysis of the SEB and AMPS data was undertaken by J.V.T, A.K and J.C.K. The author P.K.M ran the SEB model and provided discussion of the results. J.V.T, A.K and A.N.R contributed to the development of the research question. All authors

contributed to the progress of the manuscript and the discussion of the results.

**Competing Interests**

The authors declare no conflict of interest.

**Acknowledgements**

The authors would like to acknowledge the Natural Environment Research Council (NERC) UK grant NE/G014124/1 "Orographic flows and climate of the Antarctic Peninsula" and NERC funded PhD studentship NE/L501633/1. We thank the National Centre for Atmospheric Research (NCAR) USA, for access to the archived AMPS output. We also appreciate the work of IMAU, CIRES and the British Antarctic Survey staff at Rothera station for their field support.

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
