# Peer review of "The influence of föhn winds on annual and seasonal surface melt on the Larsen C Ice Shelf, Antarctica"

_The Cryosphere, 2020_

## Referee Comment (RC1) · Jan Lenaerts (Referee) · 12 Aug 2020

**Review of "The influence of föhn winds on annual and seasonal surface melt on the Larsen C Ice Shelf, Antarctica" by Jenny Turton and co-authors**

First and foremost, I would like to apologize to the authors and editor for the delay in submitting this review. Let me be clear that this delay is not a reflection on the quality of this work, but rather an oversight on my part. This is an interesting paper, and the topic definitely fits within The Cryosphere. I think the paper should go through a round of figure and table edits/improvements, changes to the introduction to give a better perspective on the goals of this study, and some clarifications on numbers stated. Please refer to detailed comments below.

L 46: The study by Datta et al. (2019, https://doi.org/10.1029/2018GL080845) seems to be missing from the introduction and overview (and the reference list), while it seems to share a lot of methods and results. It would be useful to add a discussion on similarities and differences. More generally, I find it challenging to pinpoint the originality of this paper within the realm of recent papers on foehn-driven melt. For example, how is this study different from the Kuipers Munneke et al., 2018 study (https://doi.org/10.1029/2018GL077899), who does focus on a single location but also presents melt maps? And how does this study build from the SEB study by Kuipers Munneke et al., 2012 and from Turton et al., 2018? What are some of its unique features, e.g. the methods, the data, the results?

Figure 1: It would be more logical to zoom in from subpanels a to c, instead of zoom out.

Table 2: it is unclear (1) if this is a new result or simply a repetition of Turton et al., 2018; and (2) whether the data gaps are considered when calculating the foehn frequency. I assume the percentages represent ratios between foehn episodes and total duration of non-missing data – but this is important to clarify; (3) what the agreement/overlap between AWS and AMPS is in general, i.e. what is the frequency of events in AWS only, AMPS only, and both – and why is that?

Table 3: it would be worthwhile representing the numbers in the first columns are percentages as well. What does the 31.4% and 33.7% represent exactly?

Figure 3: this figure needs to be improved considerably. The dots are difficult to see, black and red is difficult to separate, and units need to be added. Use subscripts as necessary.

Table 4: this might be nitpicky but try to use subscripts and superscripts in this table. Would it be an idea to combine Table 4 and Figure 5, to avoid table redundancy/overload?

Figure 4: color scale is not useful for this purpose (this would be a color scale for 0 in the middle, ranging from negative to positive numbers). Consider changing the color scale.

Figure 5: use consistent terminology throughout (e.g. $H_{sen}$ instead of 'sensible heat flux')

Figure 6: same here – this color scheme is not useful for this purpose.

Figure 7:mind the units in brackets.

---

## Referee Comment (RC2) · Anonymous Referee #2 · 19 Aug 2020

The influence of föhn winds on annual and seasonal surface melt on the Larsen C Ice Shelf, Antarctica
—review, 8/18/20

The manuscript by Turton et al. examines observational and model evidence of föhn wind impacts on surface processes across the Larsen C ice shelf. Aggregating model output and weather records from automated weather stations (AWS), the authors quantify the frequency of föhn events and their association with surface melt, teasing apart the underlying relationship between the two by examining surface energy balance. Differences in these variables between föhn and non-föhn periods are examine across seasons and years. Mean annual patterns are also present, which provide an overview föhn impacts on the region.

While akin in spirit to other publications examining the impact of föhn winds on the Larsen Ice Shelves, this manuscript distinguishes itself by examining a long-term observational weather record in context of model output to tease apart general relationships between föhn winds and surface melt. Other studies have tended to focus on specific events and shorter time-periods. As such, this study fills a niche in the field. Another point of novelty is the detailed evaluation of AMPS and its ability to resolve the relationship between föhn occurrence and melt across seasons and years. Identification of strengths and weaknesses in the model provides great insight than can further model development, and inform future studies using model output to examine these regional wind patterns.

The manuscript is overall well-written, logically organized, and the major conclusions of the study (namely the importance of spring föhn events for seasonal / annual melt amounts / patterns) are well supported by the data presented. There is some repetition in the results, owing to the structure of the manuscript which repeatedly highlights the importance of föhn for melts in different seasons (apart from winter), and years. However this is more a stylistic point and does not detract from the scientific findings of the paper.

I highlight below a number of technical points regarding the writing, with edits meant to clarify the text. While technical corrections are needed, the manuscript is scientifically sound and will contribute significantly to the field once published.

Technical corrections:

Throughout: review tenses — some mixture of present and past to describe the same data

L14: 100 km away
L17-18: previous attempts to quantify
L39-40: Marshall et al. (2006) proposed that a trend towards... events on the eastern side of the AP leading to increased surface melt.

L51: be localised but intense...
L52: 100 km (separate number and units here and elsewhere)
L54: Fohn frequency  is highly variable from season to season over LCIS
L77: However, these have focused on a number of....

L94: ... model driven by...
L101-: Meteorological observations spanning January 22, 2009 to December 31, 2012 (at AWS2 and AWS3) and February 19, 200 to December 31, 2012 (AWS1) were analyzed in this study. For this latter...
L103: Observations were collected...
L104: hourly values were derived
L111: In order to compare..., and following..., we compute
L116: ... as outlined in Sect. 2.2
L118: We used a previously published and validated SEB model, in conjunction with AWS data input, to compute the surface energy balance and its components at AWS2 and 3 (citations).
L119: Daily averages, derived from the SEB model's hourly output, are analyzed in this study.
L120: only a brief overview of the SEB model
L121: delete "The model is required..."
L162: Eq. (4) was used to calculate melt from AMPS data:
L166: melting the surface (i.e. when positive)
L181: (although föhn...)

L189: AP mountains (not a proper name)
L190: In certain seasons
L192: ..., and more frequently in summer closer to the foot of the mountains (AWS1; Table 2)
L194: ... and the relative humidity decreases at least 19 %
L195: Delete Table 2 summarizes... it's introduced earlier in line 191
L205: ... to the observationally-derived SEB
L207: model identified 214 melt days at AWS2, compared to 2918 for AMPS (Table 3)
L219: AMPS is therefore better able to represent the occurrence of melting... as opposed to melting on...
L241: AMPS will simulate temperatures at or near 0 ºC, leading to an overestimation of the total number of melt days. An overestimation...
L245: reduced overestimation is a bit awkward, smaller positive bias?
L246: Emelt, alongside lower air temperatures (see Kirchgaessner et al. 2019)
L252 Lwnet (Formatting)
L257: ...observations and AMPS that melting...
L258: typically report p-value (p<0.01) rather that 99% confidence level. What test was used?
L264: with over 40 six-hourly melt events...

L270: annually-averaged differences... in a subset of the observed SEB components.
-why only some? If only tau is not tested simply state that you show

SEB components, and specify in table that you omit tau
L281: this could bias... be more specific than "this"
L282: As shown in table 2 and in Turton et al. (2018), föhn days are not evenly distributed seasonally or inter annually
L286: ... conditions are small and non significant.
L299: The mean annual sensible heat values [do you mean fluxes?], observationally derived during föhn days for AWS2 and AWS3, are very similar... However, at AWS1, sensible heat fluxes are slightly smaller [than?]
L307: Hsen <- subscript
L316: ... during föhn conditions, although the differences with non-föhn conditions...
L323: ... whereby fewer clouds during föhn conditions lead to reduced downwelling flux of long wave...

L343: .. this section focuses solely on AWS2 data.... Mean annual melt from 2009-2012

L369: the largest increase in .... is associated with springtime föhn events.
L379: ... was significantly lower during föhn conditions than.... (p < 0.05; Table 5)
L393-394: you make the same point at the close of numerous paragraphs (see p15 L 333, 346, 356, 361, 403. Review the manuscript to see if some of these points could be condensed / aggregated to avoid repetition

Discussion
L483: perhaps a stylistic point, but I would avoid starting the discussion with caveats, or lists of data you did not have. I would highlight the data and interesting results in context and in relationship to your hypothesis (e.g. starting with L488 Here...), and move the discussion of data availability farther down
L502: the three AWS locations.
L503: ...the contradicting results of Kuipers...

L513: As a result, values of melt energy currently derived from AMPS cannot be trusted...

Figure 1: I'd list the panels in reverse order, c (more general) -> b -> a
Figure 2: The daily, observationally-derived melt energy Emelt at AWS2 (red) and from AMPS (black) for 2009-2012.
->Change legend to show AWS2
Figure 3:
-at AWS2 during... b) summer (DJF)...
-increase point size in figure, hard to see
-Emelt-> formatting
Figure 4:

–colorbar labels too small, hard to read
Figure 5: Are these means? I would appreciate seeing error bars on this figure

Tables
–for tables, add confidence intervals / sd to measurements
Table 2:
–caption: ... AMPS, following Turton et al. (2018)
–... for example, DJF 2012 spans December 1, 2011 to February 29, 2012)
–center AWS2 (%). Use parentheses or brackets, but be consistent in figures and tables

Table 3:
From observationally–derived data at AWS2, alongside AMPS model output interpolated to the same location... The total number of melt days... occurring with föhn and non–föhn periods are indicated for 2009–2012.
–melt amounts is too vague. Cite specific entry
–the same for both AWS and AMPS as a result of the föhn identification criteria (see Methods)

Table 4:
–superscript and subscripts on Variable
L294: Differences were not assessed for tau, derived from average SW and SWtop, owing to small sample size

Table 5: ∗ indicates... (remove The)

Table 6: Summertime daily–averaged values of ... ∗ indicates... t–test... and ∗∗ at the 99% confidence...
–formatting of Variables

Table 7: for all dates, comma after numerical day (e.g. November 23, 2009)

---

## Author Comment (AC1) · 1 Sep 2020

Author response: Reviewer 1, Jan Lenaerts

Dear Dr Lenaerts,

Thank you for your review and for your positive outlook on our manuscript. We appreciate your apology and understand that the current pandemic has caused additional strains on scientists. We provide a point-by-point response to your suggestions below, and we will mark any changes in red on the updated manuscript so that these can by reviewed by you again with ease. Once we have permission from our editor to upload the new manuscript, we will do so. The manuscript has now improved in clarity and the motivation is now stronger with your suggestions.

Many thanks,
Jenny Turton, on behalf of all authors.

L46: The study by Datta et al. (2019) is missing.
Yes, this is indeed an oversight. The original study was conducted in 2017, prior to the Datta paper, however it slipped by us when we were updating the manuscript ready for submission. We have now read it and added it to the manuscript where necessary. Upon uploading the final comments, we will highlight exactly where these changes are.

More generally, I find it challenging to pinpoint the originality of this paper within the realm of recent papers on foehn-driven melt. How does this study build on from the SEB study by Kuipers Munneke et al. 2012 and from Turton et al. 2018? What are some of its unique features e.g. the methods, the data, the results? How is this study different from the Kuipers Munneke et al. 2018 study?
Editor comment: initial notes from the editor also highlighted that we did not make the contribution to science clear enough. Hopefully these changes also satisfy this comment.
There are an increasing number of foehn-driven melt studies in the literature now. However, we are confident that our study provides some points of novelty and development in our understanding. We will ensure that this is clearer in the introduction and discussion of the updated manuscript. Furthermore, we provide additional information here in response to some of the specific studies you have highlighted.
Firstly, this study builds on that from Turton et al. 2018 by investigating the *impact* of the föhn events on the SEB of the Larsen C. In Turton et al. 2018, föhn were identified and their spatial and temporal distribution were investigated, however there was no presentation of the melting and SEB change due to the föhn events.
The Kuipers Munneke et al. (2018) study focuses on winter föhn melting specifically, and a much shorter period of time (in observations) than the current study. Whereas in the current study, we analyse all seasons and have 4 years of observations. We compare our winter föhn-melt results with those of Kuipers Munneke and do not see evidence of winter melting. Largely this is due to 1) the location of the AWSs are further away from the inlets and valleys where winter melt was observed 2) winter melt in Kuipers Munneke et al. (2018) may be specific to the particular season which was observed. Therefore, a combination of the Kuipers Munneke et al. (2018) study and the current study suggests that more work investigating the role and presence of winter-föhn melting should be conducted in the future.

The Kuipers Munneke et al. (2012) study focuses on the SEB over Larsen C in a more general or climatological sense. Föhn winds are only introduced as a case study of a particular event. Here, we specifically investigate the föhn impact and compare to non-föhn SEB periods. Our study builds on that of Kuipers Munneke et al. (2012) by looking at the case study in context of all föhn events over a four-year period. The current study uses the SEB model and AWS data from Kuipers Munneke et al. (2012). Furthermore, additional AWS data is used to run the SEB on the remnants of Larsen B ice shelf. This observationally-derived SEB model output (as opposed to reanalysis or atmospheric model output) has not previously been published and is here analysed to look at the impact of föhn on a location which is often overlooked in other studies (i.e King et al. 2015, Kuipers Munneke et al. 2018, only look at föhn on Larsen C, but not on Larsen B). Therefore, results SEB model data at AWS1 is novel.

The current study uses a longer observational record for analysing föhn impact on the Larsen C (four years) than in previous studies. The majority of previous studies have analysed a single föhn event or a single season with a few föhn events (e.g King et al. 2017, Kuipers Munneke et al 2012). We also analyse SEB data from a new location, north of Larsen C, which is also prone to föhn events, and has previously only been looked at in atmospheric model output. Many föhn-melt studies (e.g. Datta et al 2019, Kuipers Munneke et al. 2018) complement modelling with satellite images of melt extent or duration, but do not go into much or any detail on the SEB components responsible for the melt, as we do here. For these reasons, we believe that this work is original and provides additional information on the impact of föhn winds on the Larsen C ice shelf. We will ensure that the novelty and motivation for this study are highlighted in the updated manuscript, so that our developments are clearer to the reader.

Figure 1: It would be more logical to zoom in from subpanels a to c, instead of zoom out. We have now re-ordered the figure to zoom in.

Table 2: it is unclear (1) if this is a new result of simply a repetition of Turton et al. 2018; and 2) whether the data gaps are considered when calculating the foehn frequency. I assume the percentages represent ratios between foehn episodes and total duration of non-missing data- this this is important to clarify; 3) what's the agreement/overlap between AWS and AMPS is in general, i.e what is the frequency of events in AWS only, AMPS only, and both- and why is that?
Thank you for your questions, we hope to clarify your individual sections below:
1) This is a repetition of Turton et al. (2018), but presented as frequency percentages rather than absolute numbers as in the original 2018 study. We wanted to include this data so that the discussion surrounding the importance of frequency of föhn winds on the SEB is clear. We have changed the table caption to make this clearer.
2) Data gaps are considered when calculating the foehn frequency yes. We have included this in the caption now to make it clear.
3) This information was presented and discussed in Turton et al. (2018). As it is relevant here too, we have now included some of this information in the manuscript.

Table 3: it would be worthwhile representing the numbers in the first columns as percentages as well. What does the 31.4% and 33.7% represent exactly?

We are not 100% sure what Dr Lenaerts means by this comment. We think that he is referring to the first two rows (as opposed to columns). These are not written as percentages, as the values are used to calculate the percentages in other rows and are needed to put these percentages into context. For instance, the 31.4% is the percentage of föhn days on which melt is also observed. Out of a total of 86 föhn days, 27 of them experience melting, which is 31.4% of the time (from AWS2). Similarly, the 33.7% is the same calculation but for AMPS data. We are happy to review this table again should this still not be clear.

Figure 3: this figure needs to be improved considerably. The dots are difficult to see, black and red is difficult to separate, and units need to be added. Use subscripts as necessary. Thank you for your suggestions. We have now increased the size of the dots and changed the marker shape for non-föhn events to aid clarity, included units and subscripted the labels.

Table 4: this might be nitpicky but try to use subscripts and superscripts in this table. Would it be an idea to combine Table 4 and Figure 5, to avoid table redundancy/overload? The sub/superscripts were also raised by reviewer 2. However, Table 4 has now been removed, and additional panels added to Figure 5 to combine the two and remove the excess table information. Please note that the original Figure 4 and 5 have switched numbers to better reflect their position in the manuscript now that Table 4 is removed. Thank you for this suggestion, which has streamlined the manuscript.

Figure 4: color scale is not useful for this purpose (this would be a color scale for 0 in the middle, ranging from negative to positive numbers). Consider changing the color scale. We have now altered this to better reflect the data we are showing.

Figure 5: use consistent terminology throughout. Updated in the new figure, thank you.

Figure 6: same here- this color scheme is not useful for this purpose. We have now re-plotted the figure with a new colour scheme that is more useful for the data we are showing.

Figure 7: mind the units in brackets. Units now in brackets and sub/superscript used.

---

## Author Comment (AC2) · 1 Sep 2020

Author Response: Reviewer 2

Dear reviewer,

Thank you for your positive feedback and recommendations for improving the manuscript. We appreciate the detail with which you have reviewed. We have provided point-by-point responses to your technical corrections below. Your comments or concerns are in black and our responses are in red, with italics used to highlight specific changes in the manuscript. The updated manuscript will be uploaded once permission is granted by our editor, where changes will be marked in red to highlight the changes.

Many thanks,
Jenny Turton, on behalf of all authors.

Editor comment:
Inconsistent use of LCIS or Larsen C.
We have now changed all instances of LCIS to Larsen C, as we use Larsen A and Larsen B in the same format.

Throughout: review tenses- some mixture of present and past to describe the same data.
We have now reviewed the manuscript and made changes throughout the ensure consistency.

L14: 100 km away
Changed.
L17-18: previous attempts to quantify
Changed.
L39-40: Marshall et al. (2006) proposed that a trend towards… events on the eastern side of the AP leading to increased surface melt.
Thank you for this clarification. The sentence now reads:
*Marshall et al. (2006) proposed that a trend towards an increasingly positive Southern Annular Mode (SAM) index in the late 1960's led to a strengthening and southward movement of the circumpolar westerly winds. This increased the flow of air over the AP, and consequently led to an increase in the number of föhn events on the eastern side of the AP, leading to increased surface melt.*
L51: be localised but intense
Changed.
L52: 100 km (separate number and units here and elsewhere)
Changed. We have reviewed the manuscript and made all necessary changes.
L54: föhn frequency is highly variable rom season to season over LCIS
Changed, thank you.
L77: However, these have focused on a number of…
Changed.
L94: … model driven by…
Changed.

Ln 101-: Meteorological observations spanning January 22, 2009 to December 31, 2012 (at AWS2 and AWS3) and February 19, 2011 to December 31 (AWS1) were analysed in this study. For this latter…

Changed, thank you.

L103: Observations were collected…

Changed.

L104: hourly values were derived

Changed.

L111: In order to compare…, and following…, we compute

Changed, thank you.

L116: … as outlined in Sect. 2.2

Changed.

L118: We used a previously published and validated SEB model, in conjunction with AWS data input, to compute the surface energy balance and its components at AWS2 and AWS3 (citations).

Changed as suggested.

L119: Daily averages, derived from the SEB model's hourly output, are analysed in this study.

Changed, thank you.

L120: only a brief overview of the SEB model

Changed.

L121: delete 'The model is required…'

Removed.

L162: Eq. (4) was used to calculate melt from AMPS data:

Changed.

L166: melting the surface (i.e when positive)

Changed.

L181: (although föhn…)

Changed, thank you.

L189: AP mountains (not a proper name)

Changed.

L190: In certain seasons

Changed.

L192: …, and more frequently in summer closer to the foot of the mountains (AWS1; Table 2)

Changed.

L194: … and the relative humidity decreases at least 19 %

Changed, thank you.

L195: Delete Table 2 summarizes… it's introduced earlier in line 191.

Deleted.

L205: … to the observationally-derived SEB

Changed.

L207: model identified 214 melt days at AWS2, compared to 289 for AMPS (Table 3)

Changed.

L219: AMPS is therefore better able to represent the occurrence of melting… as opposed to melting on…

Changed.

L241: AMPS will simulate temperatures at or near 0 °C, leading to an overestimation of the total number of melt days. An overestimation…

Changed as suggested, thank you.

L245: reduced overestimation is a bit awkward, smaller positive bias?

Thank you for the suggestion. We used *'smaller positive bias in…'*

L246: Emelt, alongside lower air temperatures (see Kirchgaessner et al. 2019)

Changed.

L252: Formatting LWnet

Corrected.

L257: …observations and AMPS that melting…

Changed.

L258: typically report p-value (p<0.01) rather than 99% confidence level. What test was used?

We have now reported the p-value. It was the t-test, we have now made that clearer.

L264: with over 40 six-hourly melt events…

Changed.

L270: annually-averaged differences… in a subset of the observed SEB components. -Why only some? If only tau is not tested, simply state that you show SEB components, and specific in table that you omit tau.

Changes made as suggested.

L281: this could bias… be more specific than 'this'

We have now written '*the seasonal magnitude of SW↓ could bias…*'

L282: As shown in table 2 and in Turton et al. (2018), föhn days are not evenly distributed seasonally or interannually.

Changed, thank you.

L286: … conditions are small and non-significant.

Changed.

L299: The mean annual sensible heat values [do you mean fluxes?], observationally derived during föhn days for AWS2 and AWS3, are very similar… However, at AWS1, sensible heat fluxes are slightly smaller [than?].

Changes made as suggested. This section now reads as: *'The mean annual average sensible heat fluxes, observationally derived during föhn days at AWS2 and AWS3, are very similar (23.0 W m$^{-2}$ and 24.2 W m$^{-2}$ respectively) (Table 4). However, at AWS1 sensible heat fluxes are slightly smaller (19.7 W m$^{-2}$) than the other locations due to the…'*

L307: Hsen needs subscript

Changed, thank you.

L316: …during föhn conditions, although the differences with non-föhn conditions…

Changed.

L323: …whereby fewer clouds during föhn conditions lead to reduced downwelling flux of long wave…

Changed.

L343: … this section focuses solely on AWS2 data… Mean annual melt from 2009-2012

Changed, thank you.

L369: the largest increase in… is associated with springtime föhn events.

Changed.

L379: … was significantly lower during föhn conditions than… (p<0.05; Table 5)

Changed.

L393-394: you make the same point at the close of numerous paragraphs (see P15 L333,346,356,361,403). Review the manuscript to see if some of these points could be condensed/aggregated to avoid repetition.

Thank you for highlighting these repetitions. The repeated sentences have been removed from L333 and L346.These points have been condensed in L356. As L361 serves as a summary paragraph before the next section of results, we have left this in. We have removed the repeated sentence on L403 also.

Discussion

L483: Perhaps a stylistic point, but I would avoid starting the discussion with caveats, or lists of data you did not have. I would highlight the data and interesting results in context and in relationship to your hypothesis. (e.g starting with L488 Here…), and move the discussion of data availability farther down.

We have now re-structured the start of the discussion to take this into account. The start of the discussion now focuses on highlighting the new available data. The data limitations are discussed in paragraph 3 now instead.

L502: the three AWS locations.

Changed.

L503: … the contradicting results of Kuipers…

Changed.

L 513: As a result, values of melt energy currently derived from AMPS cannot be trusted…

Changed.

Figure 1: I'd list the panels in reverse order, c (more general) -> b -> a

As we have changed the order of the panels, we have subsequently updated the citation to match this. Please check the new figure to ensure that you are satisfied.

Figure 2: The daily, observationally-derived melt energy Emelt at AWS2 (red) and from AMPS (black) for 2009-2012.

Changed.

Figure 3:

-at AWS2 during… b) summer (DJF)… Changed.

- increase point size. Changed and marker shape altered for clarity.

- Emelt formatting. Changed.

Figure 4: Colorbar labels too small, hard to read.

Figure 5: Are these means? I would appreciate seeing error bars on this figure.

They are mean values yes, we have altered the figure caption to better reflect this. Please note, Figure 5 has been altered following reviewer #1s suggestion, and is now called Figure 4 in the text. Error bars were not added, as SD has been added to tables instead.

Tables:

-for tables, add confidence intervals/sd to measurements

SD has now been added, thank you.

Table 2:

-caption: …AMPS, following Turton et al. (2018). Changed.

-… for example, DJF 2012 spans December 1, 2011 to February 29, 2012). Changed.

- center AWS (%). Use parenthases or brackets, but be consistent in figures and tables.

Changed.

Table 3: From observationally-derived data at AWS2, alongside AMPS model output interpolated to the same location… The total number of melt days… occurring with föhn and non-föhn periods are indicated for 2009-2012. Changed

- Melt amounts is too vague. Cite specific entry. Specific entry given.
- The same for both AWS and AMPS as a result of the föhn identification criteria (see Methods). Changed.

Table 4:

-superscripts and subscripts on variable.

Table 4 was removed as the information was included in the original Figure 5 (now Figure 4).

-L294: Differences were not assessed for tau, derived from average SW and SWtop, owing to small sample size.

Changed, thank you. Although this information is no longer in the table caption, but in Figure 4 caption.

Table 5: * indicates... (remove The)

Changed.

Table 6: Summertime daily-averaged values of ... *indicates... t-test... and ** at the 99% confidence... Changed.

-formatting of variables. Changed.

Table 7: for all dates, comma after numerical day (e.g November 23, 2009)

Checked throughout manuscript and changed, thank you.

---

## Author Response (AR1)

Dear Editor,

Thank you for your feedback on the manuscript and the invitation to submit our revised manuscript following minor amendments below. We provide point-by-point responses here with author comments in red and italics used for specific manuscript sections, as well as a marked-up manuscript with changes highlighted in red. We appreciate the detailed reviews and editing of this paper during a difficult time.

Dr Jenny Turton, on behalf of all co-authors.

L31: perhaps include a reference to a hydrofracturing paper – _eg. Bell, Kingslake or Banwell.
Thank you, we have now included Robel and Banwell (2019). This section now reads as: '*The collapse of Larsen A and B was facilitated by a process known as hydrofracture, whereby ice is weakened due to drainage of surface melt water into crevasses and increased pressure from ponds of standing meltwater forming on the ice shelf surface (Scambos, 2002; Robel and Banwell 2019). Recently, Robel and Banwell (2019) confirmed that the rapid rate of collapse of Larsen B (only a few weeks) was caused by an anomalously large, sudden and widespread surface melt event, which triggered successive or simultaneous hydrofracturing.*'

L74: errant comma after '(fluxes directed towards the surface are defined positive), ' _
Removed now.

L88: 'aim to extend the scope' _is not really sufficient justification for the study. Please tell us plainly how this study is novel. Both reviewers raise this issue, and I noted it in my initial assessment. It is clear from the paper that the study is indeed new, but that it necessarily builds on existing work by yourself and other authors. I think this just requires better 'selling' _in your introduction, and also in your conclusions.
Thank you for this suggestion. Based upon yours and the reviewer's comments, the introduction and conclusion has been changed considerably. Please see the marked manuscript for specific sentences in the conclusion. Particular focus has been paid to the introduction and these sections have now been included towards the end of the introduction:
'*Most recently, Datta et al. (2019) analysed how the frequency of föhn winds influences snow melt, density and depth of water percolation over Larsen C using a regional climate model and remote sensing data. Studies investigating how föhn winds specifically influence the Surface Energy Budget (SEB) components, which are then responsible for melting, are less common and therefore explored here.*' ...

'*However, these studies have only focused on a number of case studies, during particular seasons with a large number of föhn winds, or for a particular location on Larsen C.  The interannual and seasonal influence of föhn winds on the SEB and melt characteristics are currently lacking, and therefore all seasons are investigated in this study. Our current understanding is largely from analysing extreme melting episodes related to föhn winds (e.g November 2010; King et al. 2017, Kuipers Munneke et al. 2012). Whether föhn winds are responsible for melting under more typical conditions, and if so which of the SEB components are influenced, are not as well understood, and are therefore explored in this study. Due to the break-up of Larsen B, observations on this ice shelf are limited, and previous studies investigating the role of föhn winds have focused on Larsen C. Here, we use a SEB model along with observations on the remnants of Larsen B (Scar Inlet) to understand the potential impact of föhn winds in this more northerly setting, for the first time.*' ...

*'In this study, we aim to extend the scope of previous studies by analysing the composite effects of föhn against non-föhn periods on the SEB and melt production for both the Larsen C and Larsen B (remnants) ice shelves, inter- and intra-annually. By doing so, we investigate the impact of föhn winds on each season with the hypothesis that the impact is highest in spring, when föhn winds are more frequent. Furthermore, we analyse observationally-derived model output from a previously unpublished dataset (on Scar Inlet) in combination with high-resolution AMPS output from 2009 to 2012, to provide a wider spatial analysis than many previous studies.'*

Table 3: Following Jan's suggestion, since you have included percentages in brackets for some rows, why not include for all? That gives us a quick idea of the % of melt days, foehn days as well as the foehn days with or without melt. Also, the final 'foehn' _in the caption is capitalised – _is this intentional?

The table has been adjusted slightly for ease of understanding and the percentages have now been added in. The capital was a typo which has now been changed. The table caption has been changed to reflect the updates. It now reads as: *'Table 3: The representation of surface melt from observation-derived data at AWS2, alongside AMPS model output interpolated to the same location. The total number of days with observations for 2009 to 2012 is 1439, which is used for the calculation of percentages for rows 2-4. The average $E_{melt}$ values are daily averages over the same period. The total number of föhn and non-föhn days are the same for both AWS and AMPS as a result of the föhn identification (Sect. 2.4), föhn conditions must be identified in both to be classified as föhn. The number of föhn and non-föhn days are the same in AWS and AMPS as this was a criterion for the detection of föhn winds in Turton et al. (2018).'*

L530: the first sentence of the conclusion is rather long. Could you cut it in half?

This has now been changed to shorten but also to answer the comments on novelty. It reads as: *'The discrimination between föhn and non-föhn conditions provides a robust understanding of the impact of föhn on components of the SEB and ultimately, surface melt. Furthermore, by assessing the more general response to föhn, as opposed to individual events, we now know the impact of particularly frequent föhn periods on the surface melt.'*

L541: can you tell us how the results of this study differ from those published previously? At the moment, this rather reads as if it's just another paper in a long line of similar studies. Given the concerns about novelty (see reviews and my comments previously, and above), I would appreciate a strong defence of why this particular paper is important.

Thank you for raising this. We have greatly improved the manuscript in this regard throughout, but particularly in the introduction, discussion and conclusions. As reviewer 2 also points out, the novelty is in both the length of the observations being used (as opposed to case studies or specific seasons) and in the analysis of all föhn conditions to understand the relationship between föhn and melt (as opposed to specific 'strong' föhn events or anonymously frequent periods). The results are therefore important, as we can now say for certain that spring-time föhn events have the largest impact on the melt by extending the melt season and increasing the energy available for melt. Previously, this was hypothesised based on a case study analysis of spring 2010 föhn events.
We also now know that spring 2010 was an anonymous springtime compared to other years due to the high number of föhn events. Interannual analysis shows us that föhn impact on the annual melt production is significantly larger in years with a high frequency of föhn events.

Furthermore, with the analysis of the AWS1 SEB, we have now observed that föhn winds directly impact the SEB on Larsen B also- which was previously only hypothesised from model output due to the collapse of the shelf prior to observations. Many föhn-melt studies (e.g. Datta et al 2019, Kuipers Munneke et al. 2018) complement modelling with satellite images of melt extent or duration, but do not go into detail about the SEB components responsible for the melt, as we do here.

Previous studies such as Bevan et al. 2018 and Weienekker et al. 2018 looked over a longer time period than our study, but only investigated spatial melt patterns and the number of föhn events per year, but did not look into the seasonal distribution of these föhn events, and therefore pinpoint specifically the föhn-induced melting. For example, if most of the föhn events in the early 2000s were during winter, these would not have had such a large impact as in another year with fewer föhn events but clustered in spring and summer.

Finally, another aspect of novelty is the detailed investigation of the success of AMPS at representing the total amount of surface melt, melt associated with föhn events and melt in different seasons (also associated with föhn events). Previous studies have shown the success and weaknesses of AMPS at identifying föhn winds from the upper atmospheric features and near-surface meteorological conditions, but not the surface melt (King et al. 2017, Turton et al. 2018, Kirchgaessner et al. 2019). This particular aspect was missing from the conclusions, so has been included now.

---

## Editor Decision (ED1)

Dear Dr Turton and co-authors,

Thank you for your patience with the extended review period of this paper. I am pleased to see a positive set of reviews, with some constructive suggestions that you have addressed in your initial response. I now request that you upload your updated manuscript including the changes detailed in your response. I also request a number of minor amendments (listed below) in addition to those suggested by the authors.

L31: perhaps include a reference to a hydrofracturing paper – eg. Bell, Kingslake or Banwell.

L74: errant comma after '(fluxes directed towards the surface are defined positive), '

L88: 'aim to extend the scope' is not really sufficient justification for the study. Please tell us plainly how this study is novel. Both reviewers raise this issue, and I noted it in my initial assessment. It is clear from the paper that the study is indeed new, but that it necessarily builds on existing work by yourself and other authors. I think this just requires better 'selling' in your introduction, and also in your conclusions.

Table 3: Following Jan's suggestion, since you have included percentages in brackets for some rows, why not include for all? That gives us a quick idea of the % of melt days, foehn days as well as the foehn days with or without melt. Also, the final 'foehn' in the caption is capitalised – is this intentional?

L530: the first sentence of the conclusion is rather long. Could you cut it in half?

L541: can you tell us how the results of this study differ from those published previously? At the moment, this rather reads as if it's just another paper in a long line of similar studies. Given the concerns about novelty (see reviews and my comments previously, and above), I would appreciate a strong defence of why this particular paper is important.

Thank you for your contribution to The Cryosphere, I look forward to reading your revised manuscript.

Kind regards,

Dr Liz Bagshaw